# LONG HISTORY SHORT-TERM MEMORY FOR LONG-TERM VIDEO PREDICTION

## ABSTRACT

While video prediction approaches have advanced considerably in recent years, learning to predict long-term future is challenging — ambiguous future or error propagation over time yield blurry predictions. To address this challenge, existing algorithms rely on extra supervision (e.g., action or object pose), motion flow learning, or adversarial training. In this paper, we propose a new recurrent unit, Long History Short-Term Memory (LH-STM). LH-STM incorporates long history states into a recurrent unit to learn longer range dependencies. To capture spatio-temporal dynamics in videos, we combined LH-STM with the Context-aware Video Prediction model (ContextVP). Our experiments on the KTH human actions and BAIR robot pushing datasets demonstrate that our approach produces not only sharper near-future predictions, but also farther into the future compared to the state-of-the-art methods.

## 1 INTRODUCTION

Learning the dynamics of an environment and predicting consequences in the future has become an important research problem. A common task is to train a model that accurately predicts pixel-level future frames conditioned on past frames. It can be utilized for intelligent agents to guide them to interact with the world, or for other video analysis tasks such as activity recognition. An important component of designing such models is how to effectively learn good spatio-temporal representations from video frames. The Convolutional Long Short-Term Memory (ConvLSTM) network (Xingjian et al., 2015) has been a popular model architecture choice for video prediction. However, recent state-of-the-art approaches produce high-quality predictions only for one or less then ten frames (Lotter et al., 2016; Villegas et al., 2017a; Byeon et al., 2018). Learning to predict long-term future video frames remains challenging due to 1) the presence of complex dynamics in high-dimensional video data, 2) prediction error propagation over time, and 3) inherent uncertainty of the future.

Many recent works (Denton & Fergus, 2018; Babaeizadeh et al., 2017; Lee et al., 2018) focus on the third issue by introducing stochastic models; this issue is a crucial challenge for long-term prediction. However, the architectures currently in use are not sufficiently powerful and efficient for long-term prediction, and this is also an important but unsolved problem. The model needs to extract important information from spatio-temporal data and retain this information longer into the future efficiently. Otherwise, its uncertainty about the future will increase even if the future is completely predictable given the past. Therefore, in this paper, we attempt to address the issue of learning complex dynamics of videos and minimizing long-term prediction error by fully observing the history.

We propose a novel modification of the ConvLSTM structure, Long History Short-Term Memory (LH-STM). LH-STM learns to interconnect history states and the current input by a History Soft-Selection Unit (HistSSel) and double memory modules. The weighted history states computed by our HistSSel units are combined with the history memory and then used to update the current states by the update memory. The proposed method brings the power of higher-order RNNs (Soltani & Jiang, 2016) to ConvLSTMs, which have been limited to simple recurrent mechanisms so far. The HistSSel unit acts as a short-cut to the history, so the gradient flow in the LSTM is improved. More powerful RNNs are likely to be necessary to solve the hard and unsolved problem of long-term video prediction, which is extremely challenging for current architectures. Moreover, by disentangling the history and update memories, our model can fully utilize the long history states.

This structure can better model long-term dependencies in sequential data. In this paper, the proposed modification is integrated into the ConvLSTM-based architectures to solve the long-term video prediction problem: Context-aware Video Prediction (ContextVP) model. The proposed models can fully leverage long-range spatio-temporal contexts in real-world videos. Our experiments on the KTH human actions and the BAIR robot pushing datasets show that our model produces sharp and realistic predictions for more frames into the future compared to recent state-of-the-art long-term video prediction methods.

## 2 RELATED WORK

**Learning Long-Term Dependencies with Recurrent Neural Networks (RNN):** While Long Short-Term Memory (LSTM) has been successful for sequence prediction, many recent approaches aim to capture longer-term dependencies in sequential data. Several works have proposed to allow dynamic recurrent state updates or to learn more complex transition functions. Chung et al. (2016) introduced the hierarchical multiscale RNN that captures a hierarchical representation of a sequence by encoding multiple time scales of temporal dependencies. Koutnik et al. (2014) modified the standard RNN to a Clockwork RNN that partitions hidden units and processes them at different clock speeds. Neil et al. (2016) introduced a new time gate that controls update intervals based on periodic patterns. Campos et al. (2017) proposed an explicit skipping module for state updates. Zilly et al. (2017) increased the recurrent transition depth with highway layers. Fast-Slow Recurrent Neural Networks (Mujika et al., 2017) incorporate ideas from both multiscale (Schmidhuber, 1992; El Hihi & Bengio, 1996; Chung et al., 2016) and deep transition (Pascanu et al., 2013; Zilly et al., 2017) RNNs. The advantages of the above approaches are efficient information propagation through time, better long memory traces, and generalization to unseen data.

Alternative solutions include the use of history states, an attention model or skip connections. Soltani & Jiang (2016) investigated a higher order RNN to aggregate more history information and showed that it is beneficial for long range sequence modeling. Cheng et al. (2016) deployed an attention mechanism in LSTM to induce relations between input and history states. Gui et al. (2018) incorporated into an LSTM, dynamic skip connections and reinforcement learning to model long-term dependencies. These approaches use the history states in a single LSTM by directly adding more recurrent connections or adding an attention module in the memory cell. These models are used for one dimensional sequence modeling, whereas our proposed approach separates the history and update memories that learn to encode the long-range relevant history states. Furthermore, our approach is more suitable for high-dimensional (*e.g.*, video) prediction tasks.

**Video Prediction:** The main issue in long-term pixel-level video prediction is how to capture long-term dynamics and handle uncertainty of the future while maintaining sharpness and realism. Oh et al. (2015) introduced action-conditioned video prediction using a Convolutional Neural Network (CNN) architecture. Villegas et al. (2017b) and Wichers et al. (2018) focused on a hierarchical model to predict long-term videos. Their model estimates high-level structure before generating pixel-level predictions. However, the approach by Villegas et al. (2017b) requires object pose information as ground truth during training. Finn et al. (2016) used ConvLSTM to explicitly model pixel motions. To generate high-quality predictions, many approaches train with an adversarial loss (Mathieu et al., 2015; Wichers et al., 2018; Vondrick et al., 2016; Vondrick & Torralba, 2017; Denton et al., 2017; Lee et al., 2018). Weissenborn et al. (2019) introduced local self-attention on videos directly for a large scale video precessing. Another active line of investigation is to train stochastic prediction models using VAEs (Denton & Fergus, 2018; Babaeizadeh et al., 2017; Lee et al., 2018). These models predict plausible futures by sampling latent variables and produce long-range future predictions.

The Spatio-temporal LSTM (Wang et al., 2017; 2018a) was introduced to better represent the dynamics of videos. This model is able to learn spatial and temporal representations simultaneously. Byeon et al. (2018) introduced a Multi-Dimensional LSTM-based approach (Stollenga et al., 2015) for video prediction. It contains directional ConvLSTM-like units that efficiently aggregate the entire spatio-temporal contextual information. Wang et al. (2018b) recently proposed a memory recall function with 3D ConvLSTM. This work is the most related to our approach. It uses a set of cell states with an attention mechanism to capture the long-term frame interaction similar to the work of Cheng et al. (2016). With 3D-convolutional operations, the model is able to capture short-term and long-term information flow. In contrast to this work, the attention mechanism in our model is

used for a set of hidden state. We also disentangle the history and update memory cells to better memorize and restore the relevant information. By integrating a double memory LH-STM into the context-aware video prediction (ContextVP) model (Byeon et al., 2018), our networks can capture the entire spatio-temporal context for a long-range video sequence.

## 3 METHOD

In this section, we first describe the standard ConvLSTM architecture and then introduce the LH-STM. Finally, we explain the ConvLSTM-based network architectures for multi-frame video prediction using the Context-aware Video Prediction model (ContextVP) (Byeon et al., 2018).

### 3.1 CONVOLUTIONAL LSTM

Let $X_1^n = \{X_1, ..., X_n\}$ be an input sequence of length $n$. $X_k \in \mathbb{R}^{h \times w \times c}$ is the $k$-th frame, where $k \in \{1, ..., n\}$, $h$ is the height, $w$ the width, and $c$ the number of channels. For the input frame $X_k$, a ConvLSTM unit computes the current cell and hidden states $(C_k, H_k)$ given the cell and hidden states from the previous frame, $(C_{k-1}, H_{k-1})$:

$$C_k, H_k = ConvLSTM(X_k, H_{k\text{-}1}, C_{k\text{-}1}), \tag{1}$$

by computing the input, forget, output gates $i_k, f_k, o_k$, and the transformed cell state $\hat{C}_k$:

$$
\begin{aligned}
i_k &= \sigma(W_i * X_k + M_i * H_{k\text{-}1} + b_i), \\
f_k &= \sigma(W_f * X_k + M_f * H_{k\text{-}1} + b_f), \\
o_k &= \sigma(W_o * X_k + M_o * H_{k\text{-}1} + b_o), \\
\hat{C}_k &= \tanh(W_{\hat{c}} * X_k + M_{\hat{c}} * H_{k\text{-}1} + b_{\hat{c}}), \\
C_k &= f_k \odot C_{k\text{-}1} + i_k \odot \hat{C}_k, \\
H_k &= o_k \odot \tanh(C_k),
\end{aligned}
\tag{2}
$$

where $\sigma$ is the sigmoid function, $W$ and $M$ are 2D convolutional kernels for input-to-state and state-to-state transitions, ($*$) is the convolution operation, and ($\odot$) is element-wise multiplication. The size of the weight matrices depends on the size of convolutional kernel and the number of hidden units.

### 3.2 LONG HISTORY SHORT-TERM MEMORY (LH-STM)

LH-STM is an extension of the standard ConvLSTM model by integrating a set of history states into the LSTM unit. The history states include the spatio-temporal context of each frame in addition to the pixel level information in the frame itself. Figure 1 illustrates the differences between a standard RNN, Higher-Order RNN (Soltani & Jiang, 2016), and our proposed model (Double and Single LH-STM).

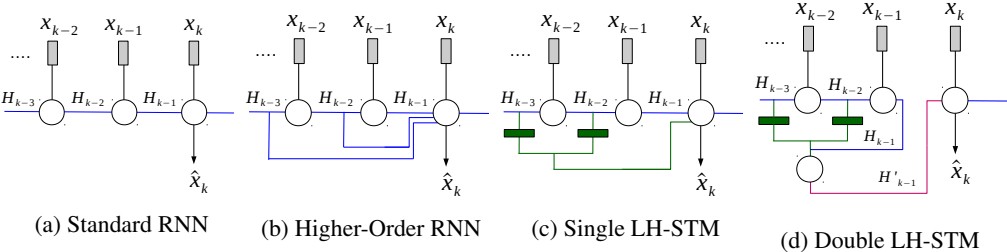

(a) Standard RNN    (b) Higher-Order RNN    (c) Single LH-STM    (d) Double LH-STM

Figure 1: Graphical illustrations of the standard RNN, Higher-Order RNN (Soltani & Jiang, 2016), Single and Double LH-STM. $X_k$ and $\hat{X}_k$, respectively indicate the input and predicted frame at time $k$. $H_{k-3}, H_{k-2}, H_{k-1}$ are past hidden states. A blue line indicates a recurrent connection; green color indicates History Soft-Selection unit, and a circle donates a memory unit.

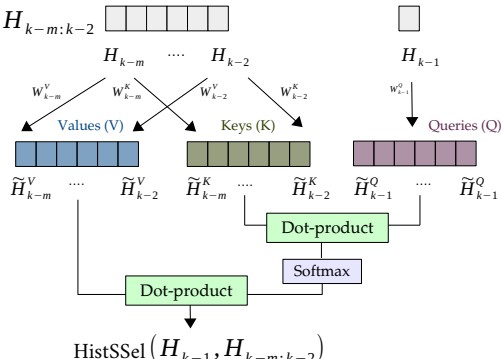

Figure 2: A graphical illustration of the History Soft-Selection Unit (HistSSel). It computes the relationship between the last $H_{k-1}$ and the earlier hidden state $H_{k-m:k-2}$.

**History Soft-Selection Unit (HistSSel):** The HistSSel unit computes the relationship between the recent history and the earlier ones using dot-product similar to (Vaswani et al., 2017)[1]. This mechanism can be formulated as $\text{SoftSel}(Q, K, V) = \text{softmax}(W^Q Q \cdot W^Q K) \cdot W^Q V$. It consists of queries (Q), keys (K) and values (V). It computes the dot products of the queries and the keys; and then applies the a softmax function. Finally, the values (V) are weighted by the outputs of softmax function. The queries, keys, and values can be optionally transformed by the $W^Q$, $W^K$, and $W^V$ matrices.

Using this mechanism, HistSSel computes the relationship between the last hidden state $H_{k\text{-}1}$ and the earlier hidden states $H_{k\text{-}m:k\text{-}2}$ at time step $k$ (See Figure 2). $H_{k\text{-}m:k\text{-}2}$ is the set of previous hidden states, $(H_{k\text{-}m}, H_{k\text{-}m\text{-}1}, \cdots H_{k\text{-}3}, H_{k\text{-}2})$. We will show in Section 4.2 the benefit of using history states versus using the past input frames directly. The history soft-selection mechanism can be formulated as follows:

$$\text{HistSSel}(H_{k\text{-}1}, H_{k\text{-}m:k\text{-}2}) = \sum_{i=2}^{n} \text{softmax}_i(\tilde{H}_{k\text{-}1}^Q \cdot \tilde{H}_{k\text{-}i}^K) \cdot \tilde{H}_{k\text{-}i}^V,$$
$$\tilde{H}_i^j = W_i^j H_i + b_i^j, \ j \in \{Q, K, V\}.^2 \tag{3}$$

**Single LH-STM:** A simple way to employ HistSSel in ConvLSTM (Equation 1) is to add $\text{HistSSel}(H_{k\text{-}1}, H_{k\text{-}m:k\text{-}2})$ in addition to the input and the previous states $(X_k, H_{k\text{-}1}, C_{k\text{-}1})$ in the Equation 6 Figure 1c shows a diagram of the computation. This direct extension is named *Single LH-STM*: $H_k = SingleConvLSTM(X_k, H_{k\text{-}1}, C_{k\text{-}1}, \text{HistSSel}(H_{k\text{-}1}, H_{k\text{-}m:k\text{-}2}))$.

$$i_k = \sigma(W_i * X_k + M_i * H_{k\text{-}1} + \text{HistSSel}(H_{k\text{-}1}, H_{k\text{-}m:k\text{-}2}) + b_i),$$
$$f_k = \sigma(W_f * X_k + M_f * H_{k\text{-}1} + \text{HistSSel}(H_{k\text{-}1}, H_{k\text{-}m:k\text{-}2}) + b_f),$$
$$o_k = \sigma(W_o * X_k + M_o * H_{k\text{-}1} + \text{HistSSel}(H_{k\text{-}1}, H_{k\text{-}m:k\text{-}2}) + b_o)$$
$$\hat{C}_k = \tanh(W_{\hat{c}} * X_k + M_{\hat{c}} * H_{k\text{-}1} + \text{HistSSel}(H_{k\text{-}1}, H_{k\text{-}m:k\text{-}2}) + b_{\hat{c}}), \tag{4}$$
$$C_k = i_k \odot \hat{C}_k + f_k \odot C_{k\text{-}1},$$
$$H_k = o_k \odot \tanh(C_k).$$

**Double LH-STM:** To effectively learn dynamics from the history states, we propose *Double LH-STM*. It contains two ConvLSTM blocks, History LSTM (H-LSTM) and Update LSTM (U-LSTM). The goal of the Double LH-STM is to explicitly separate the long-term history memory and the update memory. By disentangling these, the model can better encode complex long-range history and keep track of the their dependencies. Figure 1d illustrates a diagram of Double LH-STM.

---

[1]The purpose of this unit is to compute the importance of each history state. While we used a self-attention-like unit in this paper, this can be achieved with other common layers as well, e.g., fully connected or convolutional layers.

[2]$b_i^j$ can be omitted.

The H-LSTM block explicitly learns the complex transition function from the (possibly entire) set of past hidden states, $H_{k\text{-}m:k\text{-}1}, 1 < m < k$. If $m = k$, H-LSTM incorporates the entire history up to the time step $k - 1$.

The U-LSTM block updates the states $H_k$ and $C_k$ for the time step $k$, given the input $X_k$, previous cell state $C_{k-1}$, and the output of the H-LSTM, $H'_{k-1}$.

The History-LSTM (H-LSTM) and Update-LSTM (U-LSTM) can be formulated as:

**H-LSTM**                                                                       **U-LSTM**

$$i'_{k\text{-}1} = \sigma(M'_i * H_{k\text{-}1} + \text{HistSSel}(H_{k\text{-}1}, H_{k\text{-}m:k\text{-}2}) + b'_i), \qquad i_k = \sigma(W_i * X_k + M_i * H'_{k-1} + b_i),$$

$$f'_{k\text{-}1} = \sigma(M'_f * H_{k\text{-}1} + \text{HistSSel}(H_{k\text{-}1}, H_{k\text{-}m:k\text{-}2}) + b'_f), \qquad f_k = \sigma(W_f * X_k + M_f * H''_{k\text{-}1} + b_f),$$

$$o'_{k\text{-}1} = \sigma(M'_o * H_{k\text{-}1} + \text{HistSSel}(H_{k\text{-}1}, H_{k\text{-}m:k\text{-}2}) + b'_o) \qquad o_k = \sigma(W_o * X_k + M_o * H'_{k-1} + b_o),$$

$$\hat{C}'_{k\text{-}1} = \tanh(M'_{\hat{c}} * H_{k\text{-}1} + \text{HistSSel}(H_{k\text{-}1}, H_{k\text{-}m:k\text{-}2}) + b'_{\hat{c}}), \hat{C}_k = \tanh(W_{\hat{c}} * X_k + M_{\hat{c}} * H''_{k\text{-}1} + b_{\hat{c}}),$$

$$C'_{k\text{-}1} = f'_{k-1} \odot C'_{k\text{-}2} + i'_{k-1} \odot \hat{C}'_{k-1}, \qquad\qquad C_k = f_k \odot C_{k\text{-}1} + i_k \odot \hat{C}_k,$$

$$H'_{k\text{-}1} = o'_{k\text{-}1} \odot \tanh(c'_{k\text{-}1}). \qquad\qquad\qquad H_k = o_k \odot \tanh(C_k).$$

$$(5) \qquad\qquad\qquad\qquad\qquad\qquad\qquad (6)$$

### 3.3 Implementation Details

**Model Architectures:** ConvLSTM (Xingjian et al., 2015; Stollenga et al., 2015) is a popular building block for spatio-temporal sequence forecasting problems. In this paper, we focused on the recent state-of-the-art ConvLSTM-based architectures proposed by Byeon et al. (2018): the context-aware video prediction model (ContextVP-4). It can capture sufficient spatio-temporal context for video prediction and achieves state-of-the-art results on KTH human actions and BAIR robot pushing dataset. In this paper, the ConvLSTM block is replaced with LH-STM. Additionally, we exploit two types of skip connections to avoid the problem of vanishing gradients (Hochreiter et al., 2001) and allow long-term prediction: (a) between previous and the current recurrent layers (blue dotted line in Figure 9) and (b) across layers (green solid line in Figure 9). See Appendix B for the effectiveness of both skip connections.

ContextVP-4 (Byeon et al., 2018) consists of 4 layers of five directional Parallel Multi-Dimensional LSTM (PMD) units (Stollenga et al., 2015) along the $h+$, $h-$, $w+$, $w-$, and $t-$ recurrence directions. A PMD unit along the time direction ($t-$) is mathematically equivalent to a standard ConvLSTM (Equation 6). By using LSTM connectivities across the temporal and spatial dimensions, each processing layer covers the entire context in a video. Following Byeon et al. (2018), we also included weighted blending, directional weight sharing (DWS), and two skip connections between the layers 1 - 3 and 2 - 4 (see Figure 9). See Appendix B for details of the network architectures.

**Loss:** The loss function to train the networks is the combination of $L1$-norm, $L2$-norm, and perceptual loss (Johnson et al., 2016; Zhang et al., 2018):

$$\mathcal{L}(Y, \hat{X}) = \lambda_1 \mathcal{L}_1(Y, \hat{X}) + \lambda_2 \mathcal{L}_2(Y, \hat{X}) + \lambda_{pl} \mathcal{L}_{pl}(Y, \hat{X}), \qquad (7)$$

where $Y$ and $\hat{X}$ are the target and the predicted frames, respectively. $\lambda$ is a weight for each function. $Lp$-norm is defined as $\mathcal{L}_p(Y, \hat{X}) = ||Y - \hat{X}||_p$. $p = 1$ for $L1$-norm, and $p = 2$ for $L2$-norm. The perceptual loss computes the cosine distance between the feature maps extracted from a VGG-16 network pre-trained on the ImageNet dataset (Simonyan & Zisserman, 2014) as $\mathcal{L}_{pl}(y, \hat{x}) = 1 - \frac{1}{l} \sum_l \frac{1}{h^l \times w^l} \sum_{h^l, w^l} (\phi(y)_l \cdot \phi(\hat{x})_l)$, where $\phi(y)_l$ and $\phi(\hat{x})_l$ are the feature maps of the target and the predicted frames $Y$ and $\hat{X}$, respectively at the layer $l$. The size of the feature map at layer $l$ is $h^l \times w^l$.

## 4 Experiments

We evaluated our LH-STM model on the KTH human actions (Laptev et al., 2004) and BAIR action-free robot pushing (Ebert et al., 2017) datasets. Videos in the KTH human actions dataset consist of six human actions including walking, jogging, running, boxing, hand waving, and hand clapping. The BAIR action-free robot pushing dataset includes randomly moving robotic arms, which push objects on a table. The videos have a resolution of $64 \times 64$ pixels. We show quantitative comparisons

by computing frame-wise Structural Similarity (SSIM) (Wang et al., 2004), Peak Signal-to-Noise Ratio (PSNR), Visual Information Fidelity (VIF) (Sheikh & Bovik, 2005), and LPIPS (Zhang et al., 2018) between the ground truth and generated video sequences.

## 4.1 KTH HUMAN ACTIONS DATASET

We follow the experimental setup of Lee et al. (2018). We evaluate our methods on $128 \times 128$ pixels for 40 frame prediction (Table 2) We train all models to predict the next 10 frames, given 10 prior ones. For testing, all models recursively predict 40 frames.

Our Double LH-STM outperforms all competing methods on all metrics including the base model (ContextVP). Our Double LH-STM model consistently outperforms other methods for the 40 predicted frames. The improvement is larger for longer-term frame prediction into the future (40 frames). Qualitative results can be found in Figure 7 and Appendix E (Figure 14 and Figure 15). These results show that motion and human shape predicted by Double LH-STM are the closest to the ground truth.

Our models are compared with FRNN (Oliu et al., 2018), PredRNN (Wang et al., 2017), PredRNN++ (Wang et al., 2018a), E3D-LSTM (Wang et al., 2018b), SAVP-VAE and SAVP-Deterministic models (Lee et al., 2018)[3]. Our model outperforms all other methods.

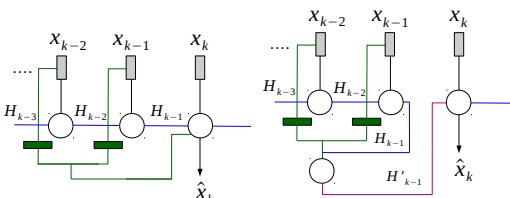

| History | LH-STM | PSNR | SSIM | VIF |
|---------|--------|------|------|-----|
| Input | Single | 25.78 | 0.779 | 0.302 |
|  | Double | 26.45 | 0.788 | 0.308 |
| State | Single | 27.83 | 0.816 | 0.354 |
|  | Double | **28.64** | **0.826** | **0.367** |

Figure 3: The use of input history with LH-STM Single (left) and Double (right)

Table 1: Comparison of using input vs state history with LH-STM Single and Double on the KTH human actions dataset (resolution $128 \times 128$).

## 4.2 STATE HISTORY VS INPUT HISTORY

The proposed LH-STM models use past hidden states. Alternatively, input frames can be directly applied to LH-STM as illustrated in Figure 3. We compared the performance of using input and state history with the same LH-STM models and the same model size in Table 1 on the KTH human actions dataset. Figure 7 shows a qualitative comparison of using state history (Double LH-STM) and input history (Double, input history). The results show the importance of using history states with LH-STM and that the state history is more suitable for videos.

## 4.3 LONGER PREDICTION

To test the performance of longer-term prediction (more than 40 frames), we additionally trained the KTH human actions dataset with a resolution of $64 \times 64$ pixels[4]. Figure 4 and Figure 5 compares the performance of 80 frame prediction with Double LH-STM and two state-of-the-art models ([3]SAVP-VAE and SAVP-Deterministic (Lee et al., 2018)) on all action classes and three action classes only (hand waving, hand clapping, and boxing), respectively. We compare only these three action classes with all action classes to show that the high performance at longer frames is not due to background copy. The human in other action class videos (running, jogging, and walking) may disappear after certain frames. Additionally, Figure 13 shows the performance comparisons of each action class. The experimental setup is the same as the experiments of a resolution $128 \times 128$ (Section 4.1). From these comparisons, we can observe that Double LH-STM results are generally better than

---

[3] We note that Lee et al. (2018) reported the performance of VAE+GAN, VAE only, GAN only, and of their deterministic models. We select their best (VAE only) and the most comparable (deterministic) models to compare against. To evaluate the VAE model, we take 100 random input samples for each video following Lee et al. (2018). The reported numbers in this paper are for their method's "median" output among the 100 randomly generated ones, compared to the ground truth. Please see Appendix C for the discussion of this evaluation setup

[4] Due to lack of GPU memory, we could not use the original resolution for 80 frame prediction.

SAVP-Deterministic and -VAE on all metrics. These also show that our model is not just copying the background when predicting more than 40 frames.

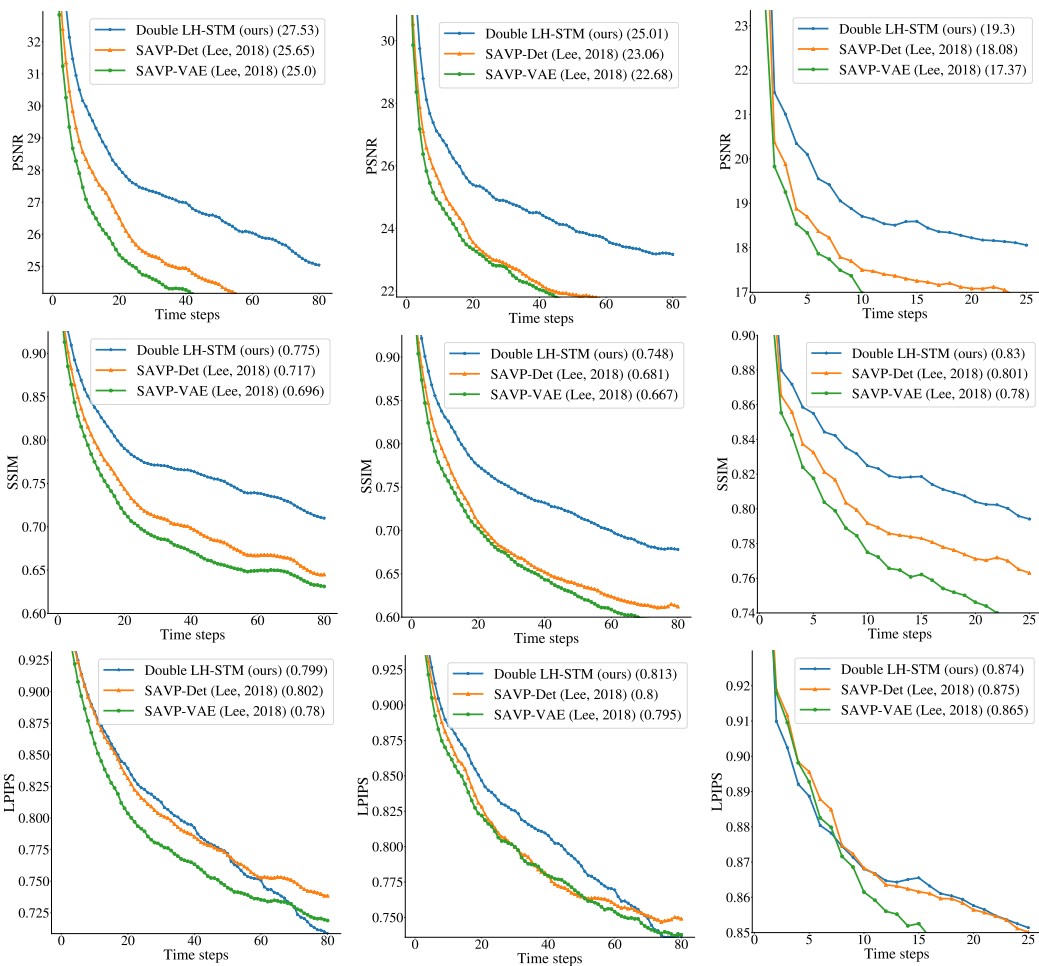

Figure 4: Per-frame comparisons on the KTH human actions dataset (all actions, $64 \times 64$, 80 frame prediction). ($\cdot$): averaged score

Figure 5: Per-frame comparisons on the KTH human actions dataset ('boxing', 'waving', and 'clapping' actions, $64 \times 64$, 80 frame prediction). ($\cdot$): averaged score

Figure 6: Per-frame comparisons on BAIR robot pushing dataset (25 frame prediction) ($\cdot$): averaged score

## 4.4 BAIR ROBOT PUSHING DATASET

Our experimental setup follows that of Lee et al. (2018). All models are conditioned on five frames to predict the next ten frames [5]. The models are tested to predict 25 frames. We evaluated our LH-STM models (Single and Double) and compared them with the base model (ContextVP), the [3]SAVP-VAE, and SAVP-Deterministic models (Lee et al., 2018) in Table 3. Our Double LH-STM achieved better PSNR and SSIM scores than the SAVP-Deterministic and SAVP-VAE models. It can be seen from the per-frame prediction results (Figure 6), that the performance LH-STM is higher over time, especially on the PSNR and SSIM metrics. On the LPIPS metric, the performance of Double LH-STM is comparable to SAVP-Deterministic and better than SAVP-VAE. Qualitative results can be found in Appendix E (Figure 17 and Figure 18). This dataset contains random robot motion which makes long-term prediction extremely hard with deterministic models. However, the overall motion prediction with Double LH-STM and SAVP-Deterministic is still reasonable compared to SAVP-VAE.

---

[5] Lee et al. (2018) used two input frames. We increase the number of input frames as motion in two frames is almost not visible. We increased the number of input frames to provide enough history to the model. We also re-trained the models of Lee et al. (2018) with five input frames.

Input        Predicted Frames (top: Ground truth)

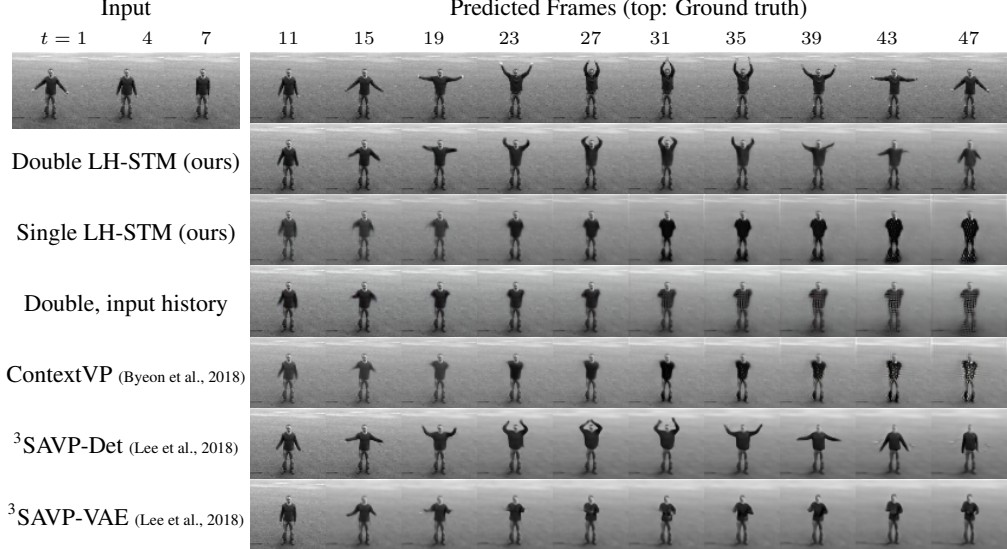

Figure 7: Qualitative comparison of Double LH-STM, Single LH-STM, Double LH-STM with input history (See Section 4.2 for the details), the base model (ContextVP (Byeon et al., 2018)), and the state-of-the-art model (SAVP-VAE and SAVP-Deterministic (Lee et al., 2018)). Double LH-STM in overall produces the sharper images and greater realism compared to other models. Compare to the Deterministic model (Lee et al., 2018), our model can capture more precise motion.

Table 2: Evaluation of multi-frame prediction on the KTH human actions dataset (resolution $128 \times 128$). All models are trained for 10 frame prediction given 10 input frames. They are then tested to recursively predict 40 future frames. The reported metrics are averages across the predicted frames. Higher values indicate better results. (*) We re-trained the model using the provided code by Wang et al. (2018a) as their originally reported numbers are not with the same experimental setup as ours. See Appendix F for training details.

| Method | PSNR | SSIM | VIF | # parameters |
|---|---|---|---|---|
| ConvLSTM (Xingjian et al., 2015) | 22.85 | 0.639 | - | - |
| FRNN (Oliu et al., 2018) | 23.77 | 0.678 | - | - |
| PredRNN (Wang et al., 2017) | 24.16 | 0.703 | - | - |
| PredRNN++ (original) (Wang et al., 2018a) | 25.21 | 0.741 | - | 15.1M |
| *PredRNN++ (retrained) (Wang et al., 2018a) | 26.37 | 0.734 | - | 15.1M |
| [3]SAVP-VAE (retrained) (Lee et al., 2018) | 26.23 | 0.777 | 0.289 | 9.8M |
| [3]SAVP-Det (retrained) (Lee et al., 2018) | 27.59 | 0.796 | 0.328 | 8.9M |
| E3D-LSTM (Wang et al., 2018b) | 27.24 | 0.810 | - | - |
| ContextVP-4 (base model) (Byeon et al., 2018) | 26.37 | 0.785 | 0.320 | 17.3M |
| Single LH-STM (ours) | 27.83 | 0.816 | 0.354 | 18.6M |
| Double LH-STM (ours) | **28.64** | **0.826** | **0.367** | 16.2M |

## 5   CONCLUSION

In this paper, we have introduced a new RNN structure, LH-STM, which incorporates long history states into the recurrent unit to learn longer range dependencies. This module is combined with the ContextVP architecture allowing context aware long-term video prediction. The resulting model produces more realistic predictions for a longer period of time compared to the recent state-of-the-art methods. The proposed approach can easily be extended to standard LSTM or Gated Recurrent Units (GRU) and applied to tasks that require a long-term memory trace, e.g., language modeling or speech recognition.

Table 3: Evaluation of multi-frame prediction on the BAIR robot pushing dataset. All models are trained for 10 frame prediction given 5 input frames. They are then tested to predict 25 future frames. The reported numbers are averages across the predicted frames. Higher values indicate better results.

| Method | PSNR | SSIM | VIF | # parameters |
|---|---|---|---|---|
| [3]SAVP-VAE (Lee et al., 2018) | 17.32 | 0.779 | 0.524 | 6.7M |
| [3]SAVP-Deterministic (Lee et al., 2018) | 18.08 | 0.801 | **0.544** | 6.4M |
| ContextVP-4 (Byeon et al., 2018) | 18.24 | 0.817 | 0.513 | 17.3M |
| Single LH-STM (ours) | 18.64 | 0.817 | 0.523 | 17.6M |
| Double LH-STM (ours) | **18.82** | **0.825** | 0.547 | 16.3M |

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

## A    NETWORK DETAILS

Each layer of ContextVP-4 contains five Parallel Multi-Dimensional LSTM (PMD) units (see Figure 8). The parameters of opposite directions are shared. We followed the same network architecture as Byeon et al. (2018) except for the number of neurons.

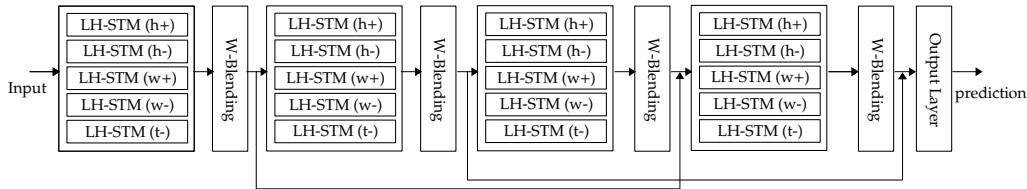

Figure 8: ContextVP-4 network archtecture

**Weighted Blending:** This layer learns the relative importance of each direction during training.

$$S_k = [H_k^{h+}, H_k^{h-}, H_k^{w+}, H_k^{w-}, H_k^{t-}],$$
$$M_k = f(W_k \cdot S_k + b_k), \qquad (8)$$

where $H$ is the output of the PMD units from $h+$, $h-$, $w+$, $w-$, and $t-$ directions, and $W$ is the weight matrix.

**Directional Weight-Sharing:** Weights and biases of the PMD units in opposite directions (*i.e.*, the $h+$ and $h-$ directions, $w+$ and $w-$ directions) are shared.

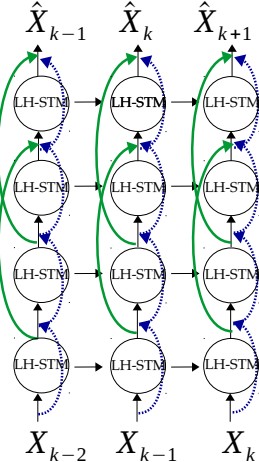

Figure 9: A graphical illustration of skip connections for the direction $t-$. Blue dotted lines indicate concatenation-based skip connection in the LH-STM block. A green solid line indicates a gated skip connection across the 1-3 and 2-4 layers.

**Skip connection in a recurrent block (R-Skip):** A skip connection is commonly used in LSTM layers (Melis et al., 2017; Merity et al., 2017; Wang et al., 2018a) for long-term gradient propagation. Our proposed model concatenates outputs of the previous and current LSTM layers (see Figure 9).

**Gated skip connection across layers (L-Skip):** Unlike in (Byeon et al., 2018), We added the gated skip connection across layers. In our model, the multiplicative gate is added to control the flow of information across layers. It stabilizes learning as depth increases. A layer is formulated as:

$$Y = H_l \cdot T_l + X_{l-i} \cdot (1 - T_l), \ i < l, \qquad (9)$$

where $H_l = X_l \cdot W_l^H + b^H$ and $T_l = \sigma(X_l \cdot W_l^T + b^T)$. $X_l$ is the input at the $l^{th}$ layer, $T$ is a gate, and $H_l$ is the transformation of the input at the layer $l$.

We adjusted the number of neurons, so all models have similar model size ($\approx$ 17M). LH-STM Double contains 64-64-128-128 neurons for the input size of $64 \times 64$ pixels and 64-128-128-128 for $128 \times 128$. LH-STM Single contains 128-128-128-128 neurons for the input size of $64 \times 64$ pixels and 64-128-128-256 for $128 \times 128$.

The initial learning rate is $1e - 4$ found by grid search between $1e - 3$ and $1e - 5$. The learning rate is decayed every 10 epochs with the decay rate 0.95. Weights are initialized using the Xavier's normalized initializer (Glorot & Bengio, 2010) and the recurrent states are all initialized to zero.

## B    ABLATION STUDY

We performed a series of ablation studies to evaluate the importance of each component of our proposed model 1) RNN type (standard ConvLSTM, Single LH-STM, or Double LH-STM); 2) different losses ($\mathcal{L}_1$, $\mathcal{L}_2$, and the perceptual loss); adding skip connections 3) in recurrent blocks; and 4) across layers (see Section 3.3 for details). We evaluated performance on the KTH human actions dataset with a resolution of $64 \times 64$ pixels. The weights for the loss, $\lambda_1$, $\lambda_2$, and $\lambda_{pl}$, are tested between zero to one. The results of this experiment are summarized in Table 4. We found that performance improves by adding skip connections in recurrent blocks and across layers. Double LH-STM outperforms the standard ConvLSTM and Single LH-STM. Combination of $\mathcal{L}_1$, $\mathcal{L}_2$, and $\mathcal{L}_{pl}$ also improves the performance (with $\lambda_1 = 1$, $\lambda_2 = 1$, $\lambda_{pl} = 1$).

## C    EVALUATION OF STOCHASTIC MODELS

SAVP-VAE numbers in the paper by  Lee et al. (2018) are obtained by first generating 100 output samples per input and then selecting the best generated sample for each input based on the highest score between the output and the ground truth. This evaluation measures the best prediction the model can do, given 100 tries. However, it does not provide an overall picture of the generations from the model. In real world scenarios, the ground truth is not available, so the 'best' samples cannot be picked as done in such evaluations. Therefore, we re-computed the scores for stochastic models based on their 'median' output sample (instead of best) among the 100 randomly generated ones, compared to the ground truth. This strategy measures how well the model can be expected to perform on average, and so it is a more representative score.

Table 4: An ablation study on KTH human actions dataset. The frame resolution is $64 \times 64$ pixels. All models are trained to predict 10 future frames given 10 input frames. They are then tested to recursively predict 40 future frames. The reported numbers are averages over all the predicted frames.

| RNN type | Loss | R-Skip | L-Skip | PSNR | SSIM |
|---|---|---|---|---|---|
| | $\mathcal{L}_1$ | | | 27.74 | 0.812 |
| ConvLSTM | $\mathcal{L}_1$ | | ✓ | 27.96 | 0.812 |
| | $\mathcal{L}_1$ | ✓ | ✓ | 28.41 | 0.812 |
| Single LH-STM | $\mathcal{L}_1$ | ✓ | ✓ | 28.59 | 0.818 |
| | $\mathcal{L}_1+\mathcal{L}_{pl}$ | ✓ | ✓ | 28.46 | 0.817 |
| | $\mathcal{L}_1$ | ✓ | ✓ | 28.69 | 0.812 |
| Double LH-STM | $\mathcal{L}_1+\mathcal{L}_{pl}$ | ✓ | ✓ | 28.20 | 0.811 |
| | $\mathcal{L}_1+\mathcal{L}_2$ | ✓ | ✓ | 29.01 | 0.818 |
| | $\mathcal{L}_1+\mathcal{L}_2+\mathcal{L}_{pl}$ | ✓ | ✓ | **29.05** | **0.820** |

## D   EVALUATION WITH LARGE AND SMALL MOTION

We split the video samples into five bins by motion magnitude computed as averaged L2 norm between target frames similar to the evaluations by Villegas et al. (2017a). Figure 12 shows the comparisons on the KTH human action dataset, and **??** shows on BAIR action free robot pushing dataset. Overall, across all metrics, all models perform worse when the motion is larger. For PSNR and SSIM, Double LH-STM achieves the best performance except for the largest motion. For LPIPS, Double LH-STM performs better for small motions (first two bins). SAVP-Deterministic performs the best on this metric for larger motions.

## E   QUALITATIVE RESULTS

Figure 14 and Figure 15 show prediction results on the KTH human actions dataset, and Figure 17 and Figure 18 presents the results on the BAIR action free robot pushing dataset. Figure 16 and Figure 19 include the samples with one of the largest motions for both datasets. When motion is larger, all models fail to produce reasonable predictions, whether they are deterministic or stochastic.

## F   RETRAINING PREDRNN++

As mentioned in the paper, we re-trained PredRNN++ on the KTH human actions dataset using the provided code by Wang et al. (2018a) (`https://github.com/Yunbo426/predrnn-pp`). We conducted grid hyper-parameter search for the learning rate and the patch size. We found in the code that there is an option to divide the input frames into smaller patches. The learning rate search was performed in the range between $1e-3$ and $1e-4$. The range of patch size was 1, 2, 4 for both frame resolutions ($64 \times 64$ and $128 \times 128$ pixels). The reported results in the paper are for a learning rate of $1e-4$ and the patch size 1 (full resolution).

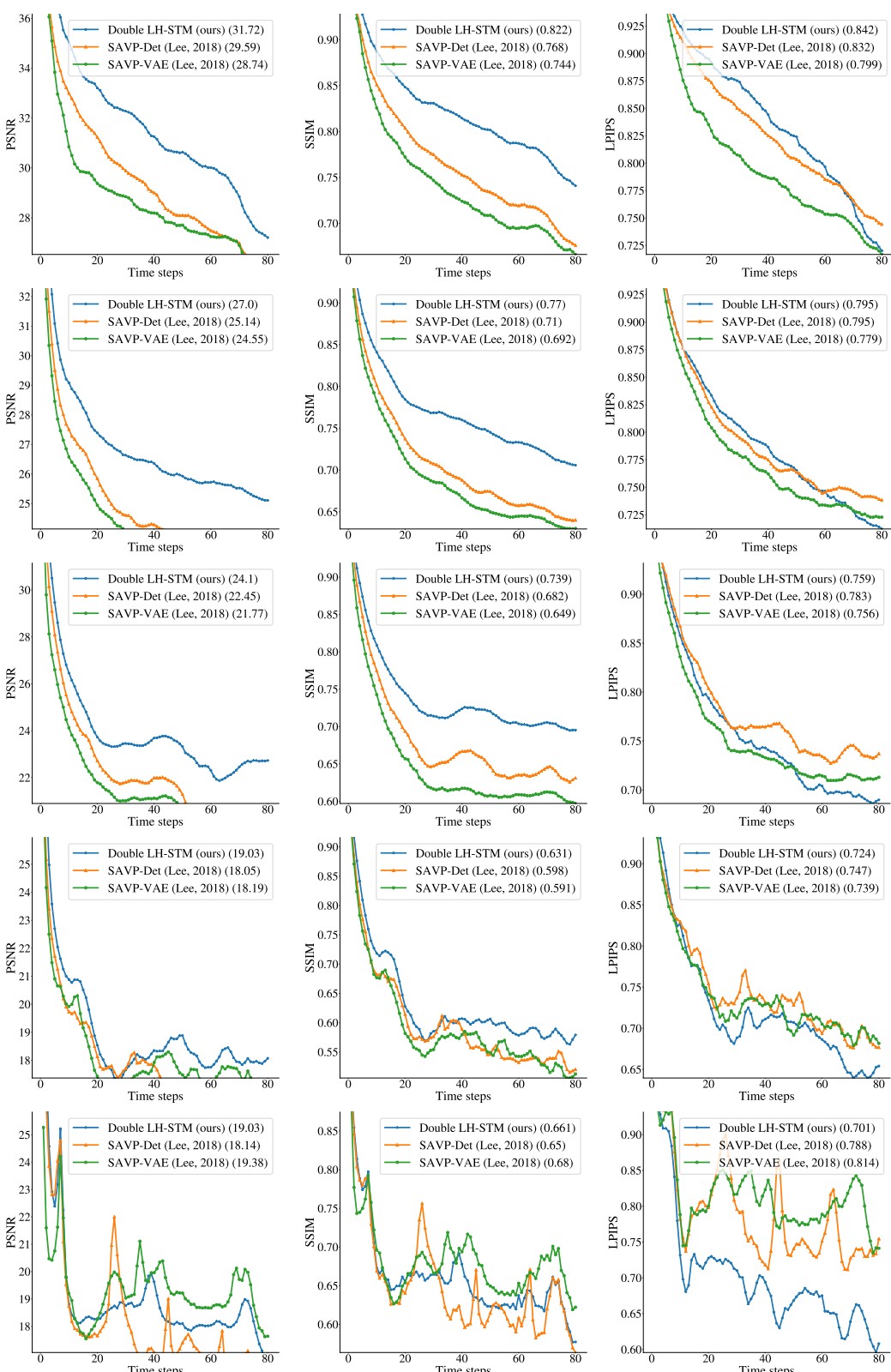

Figure 10: Per-frame comparisons of KTH human actions dataset (80 frame prediction, resolution $64\times64$) with motion-based test set separation. We split the video samples into five bins by motion magnitude computed as averaged $\mathcal{L}_2$ norm between target frames following Villegas et al. (2017a). The first row is with smallest motion and the bottom row with largest motion. ($\cdot$) indicates averaged score.

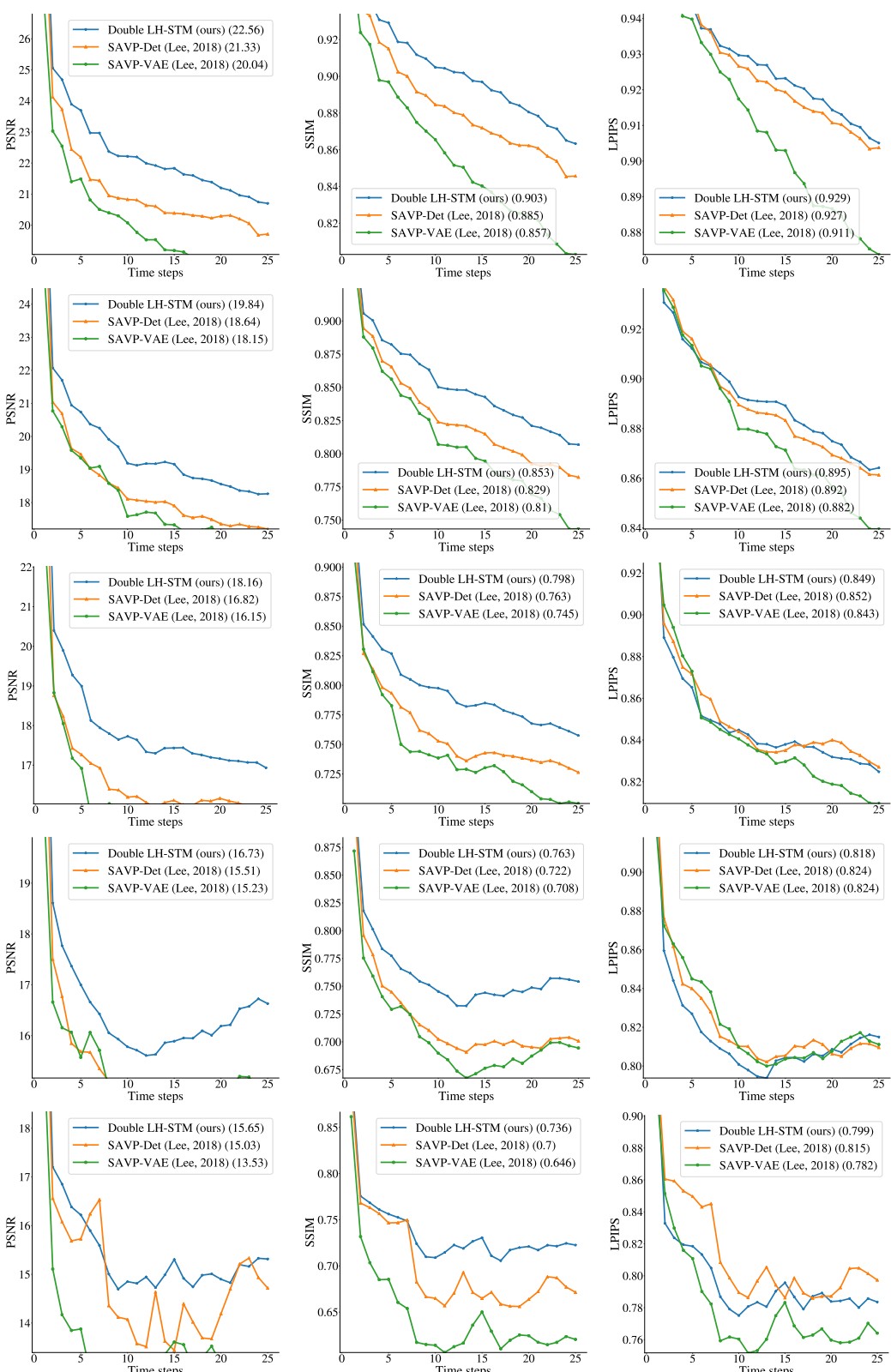

Figure 11: Per-frame comparisons of BAIR robot pushing dataset with motion-based test set separation. We split the video samples into five bins by motion magnitude computed as averaged $\mathcal{L}_2$ norm between target frames following Villegas et al. (2017a). The first row is with smallest motion and the bottom row with largest motion. ($\cdot$) indicates averaged score.

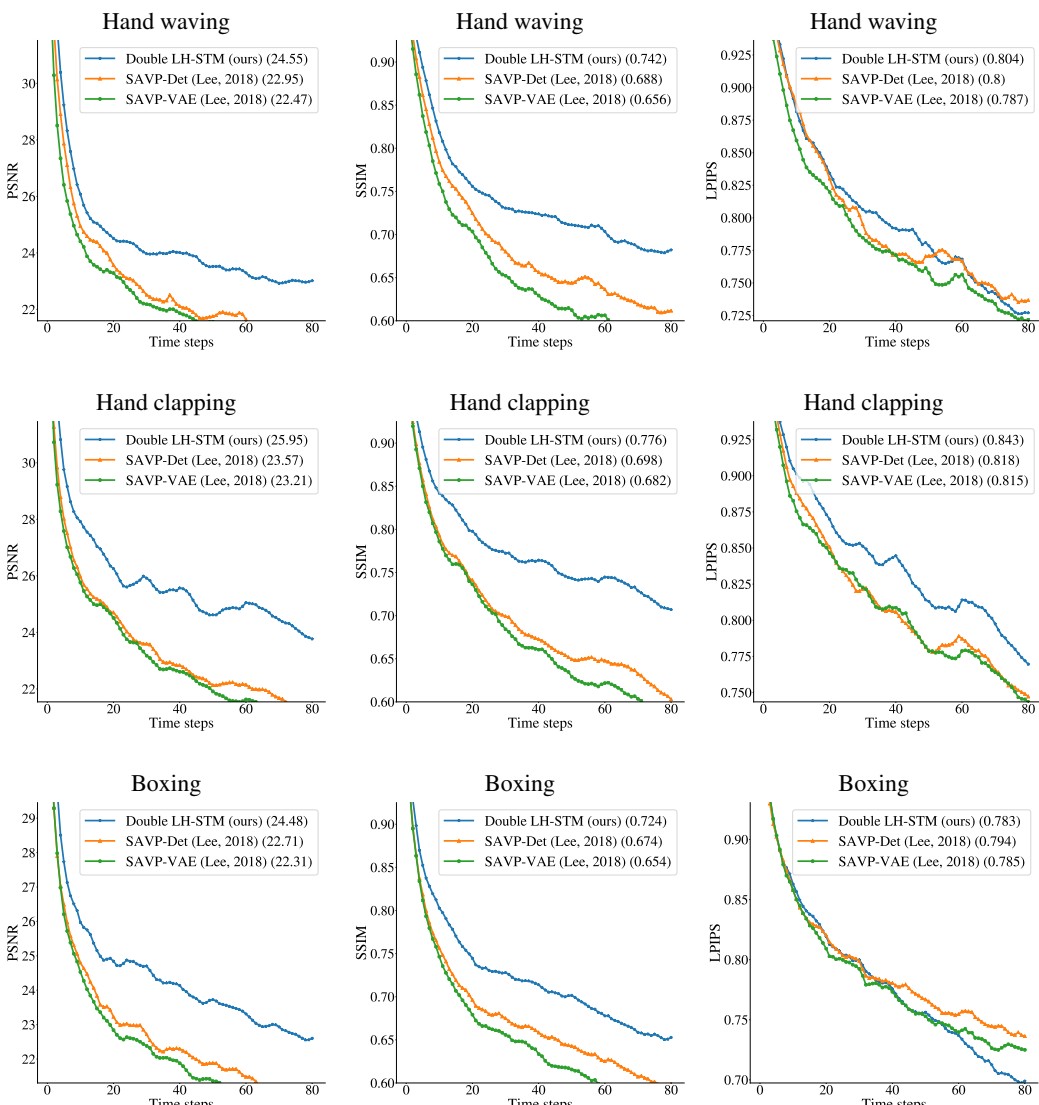

Figure 12: 80 frame prediction on KTH human actions dataset separated by actions: hand waving, hand clapping, and boxing. (·) indicates averaged score.

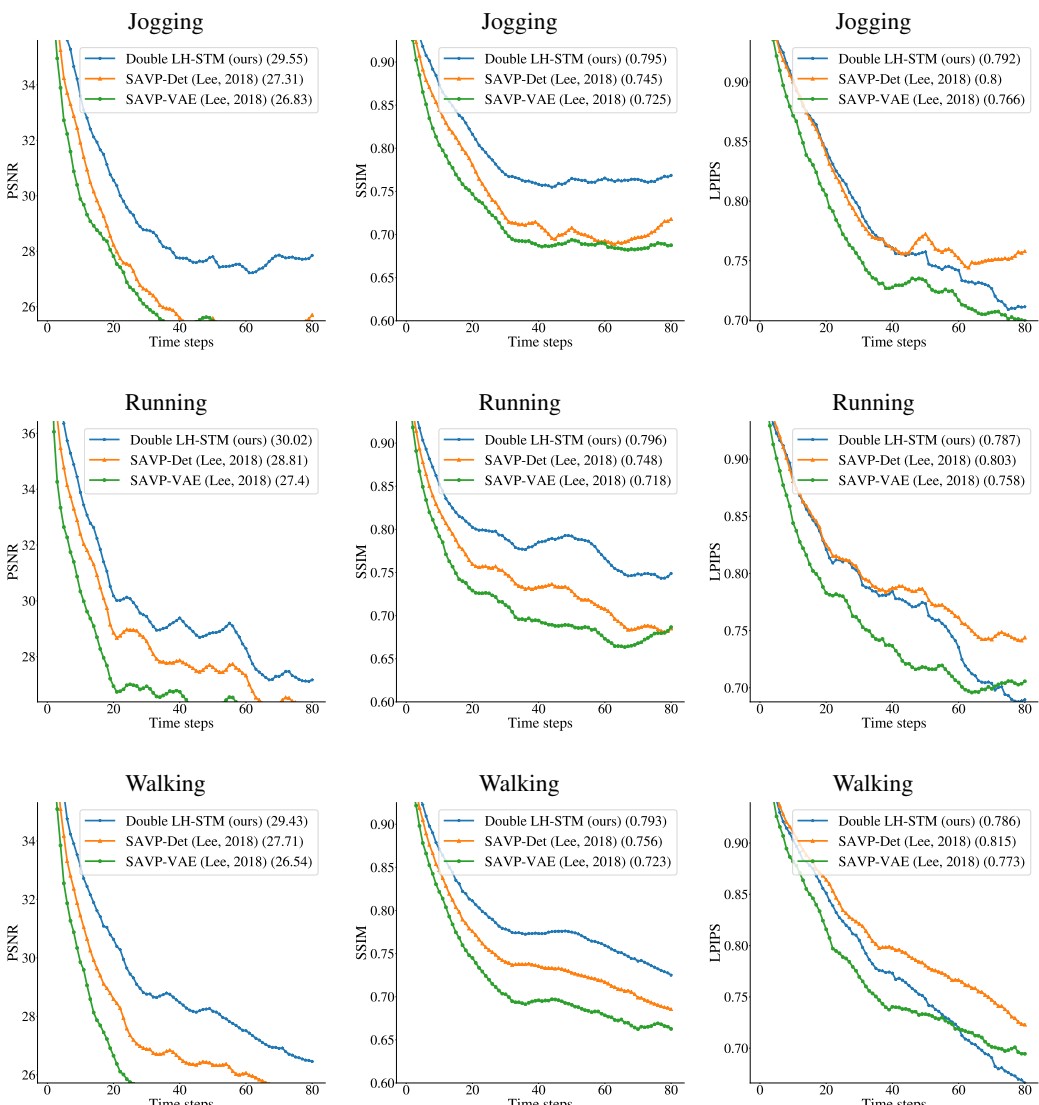

Figure 13: 80 frame prediction on KTH human actions dataset separated by actions: jogging, running, and walking. (·) indicates averaged score.

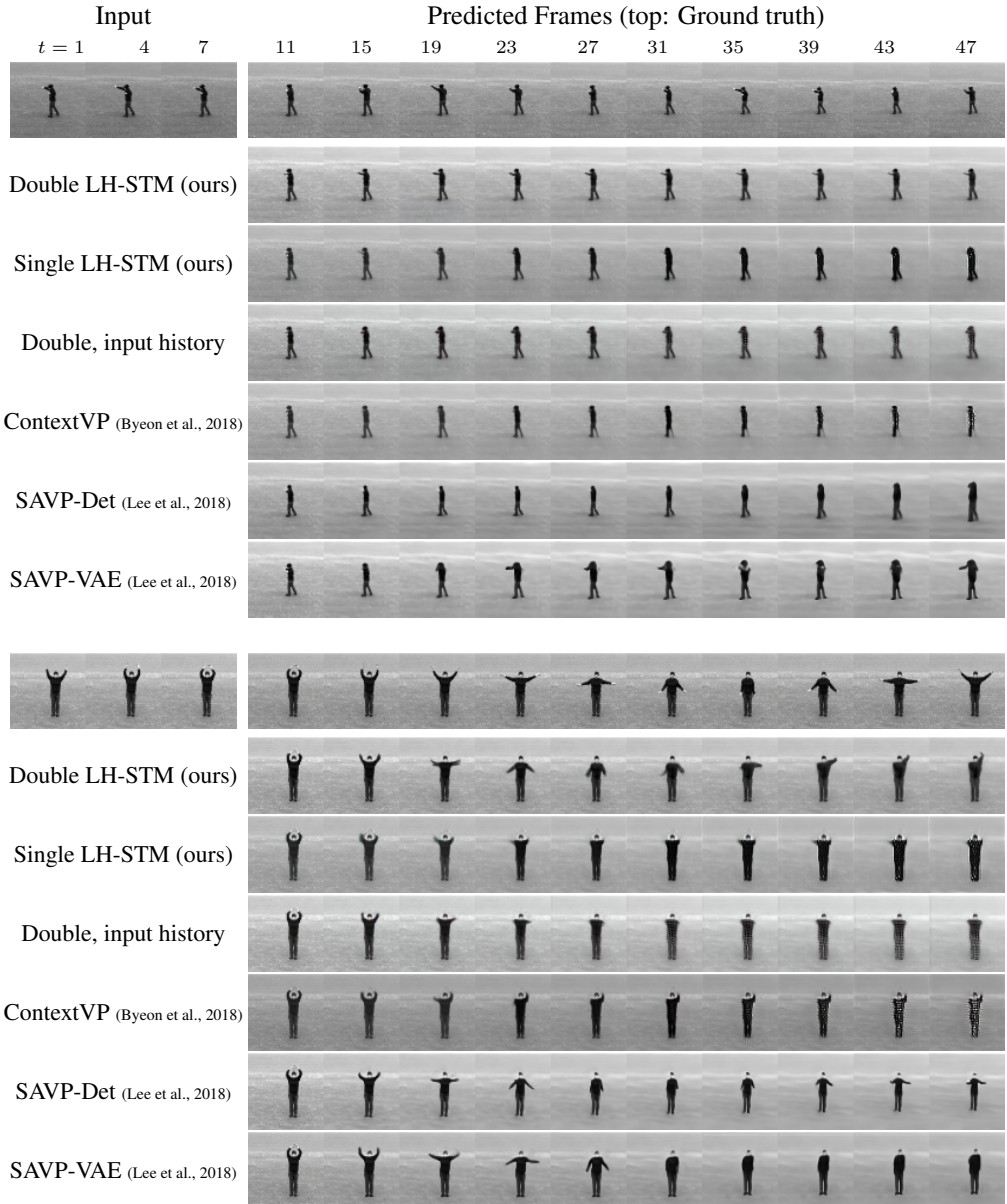

Figure 14: Qualitative comparison of Double LH-STM, Single LH-STM, Double LH-STM with input history (See Section 4.2 for the details), the base model (ContextVP (Byeon et al., 2018)), and the state-of-the-art models (SAVP-Deterministic and SAVP-VAE Lee et al. (2018)) on the KTH human actions dataset. Top row shows the ground truth, and the rest is the prediction results. The model is trained for 10 frames and predicted for 40 frames given 10 frames.

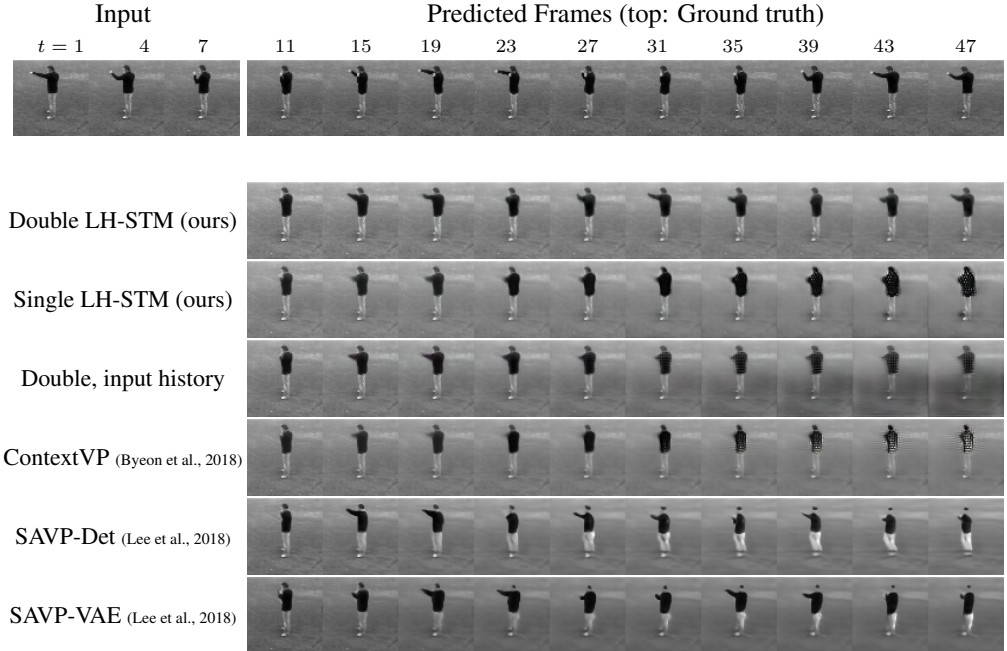

Figure 15: Qualitative comparison of Double LH-STM, Single LH-STM, Double LH-STM with input history (See Section 4.2 for the details), the base model (ContextVP (Byeon et al., 2018)), and the state-of-the-art models (SAVP-Deterministic and SAVP-VAE Lee et al. (2018)) on the KTH human actions dataset. Top row shows the ground truth, and the rest is the prediction results. The model is trained for 10 frames and predicted for 40 frames given 10 frames.

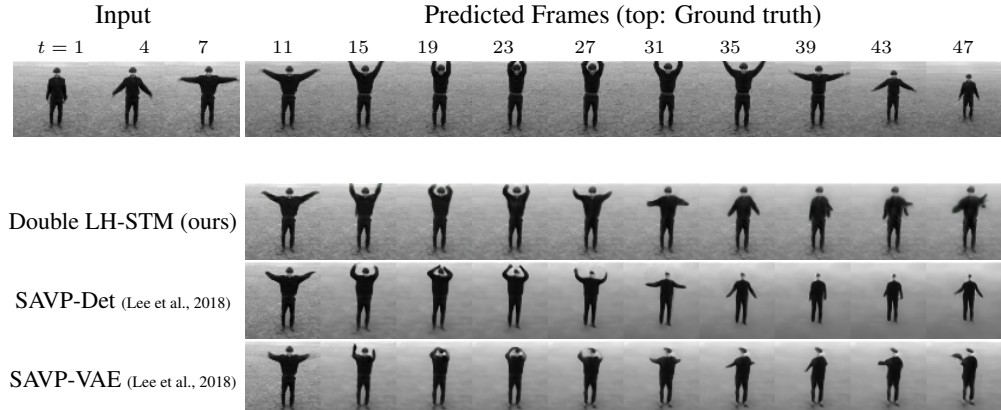

Figure 16: An example video with one of the largest motion on the KTH human actions dataset. The motion magnitude is computed as averaged $\mathcal{L}_2$ norm between target frames following Villegas et al. (2017a). Top row shows the ground truth, and the rest is the prediction results. The model is trained for 10 frames and predicted for 40 frames given 10 frames. When motion is larger, all models fail to produce reasonable predictions.

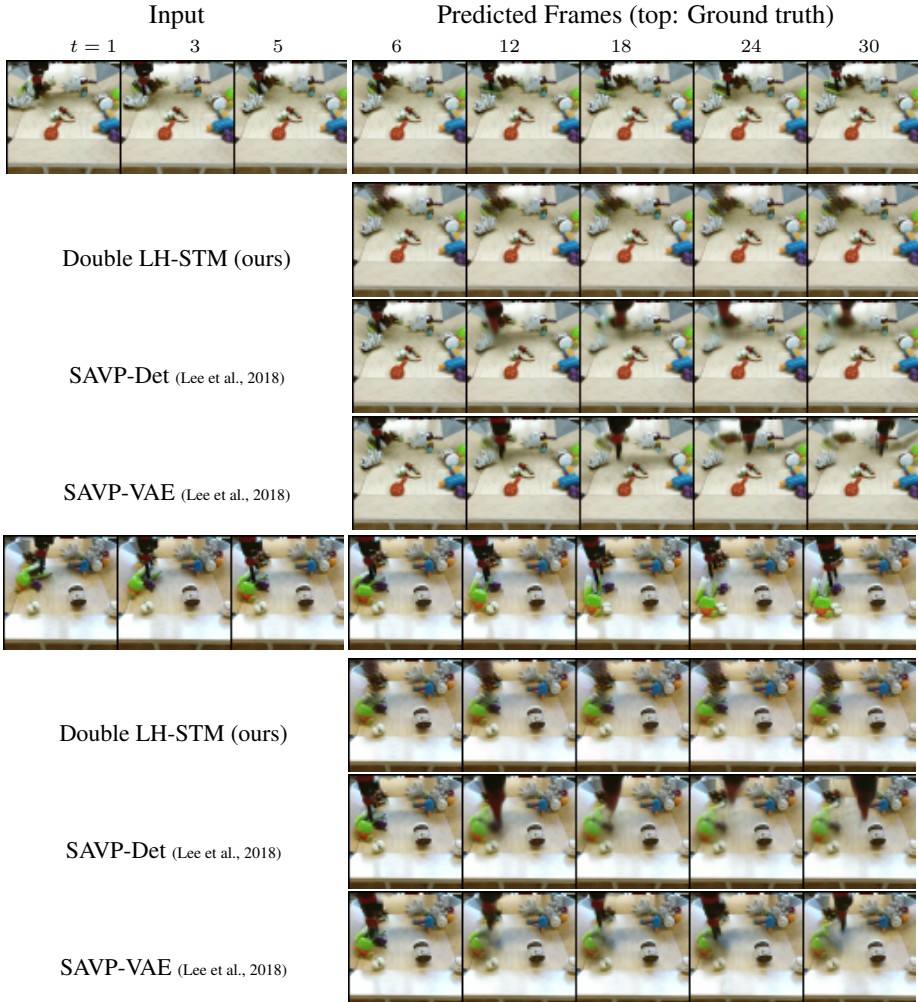

Figure 17: Qualitative comparison of Double LH-STM and the state-of-the-art models (SAVP-Deterministic and SAVP-VAE Lee et al. (2018)) on the BAIR robot pushing dataset. Top row shows the ground truth, and the rest is the prediction results. The model is trained for 10 frames and predicted for 25 frames given 5 frames.

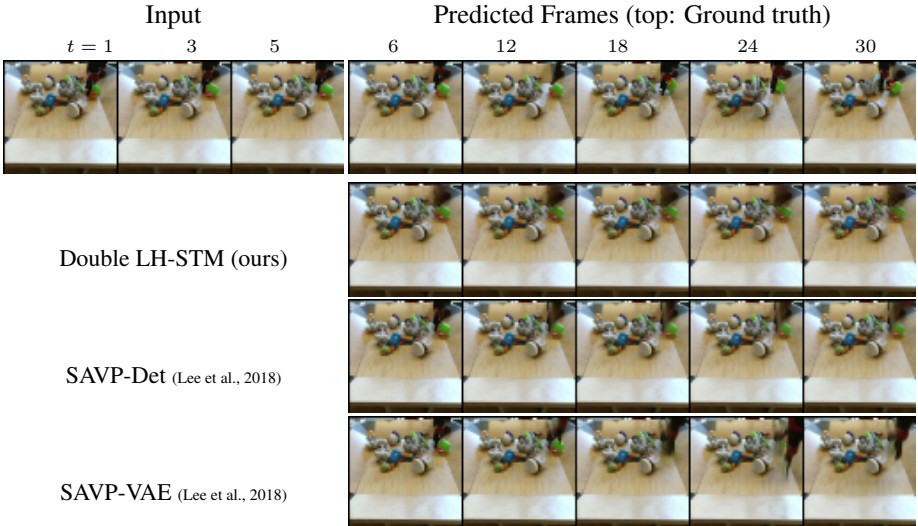

Figure 18: Qualitative comparison of Double LH-STM and the state-of-the-art models (SAVP-Deterministic and SAVP-VAE Lee et al. (2018)) on the BAIR robot pushing dataset. Top row shows the ground truth, and the rest is the prediction results. The model is trained for 10 frames and predicted for 25 frames given 5 frames.

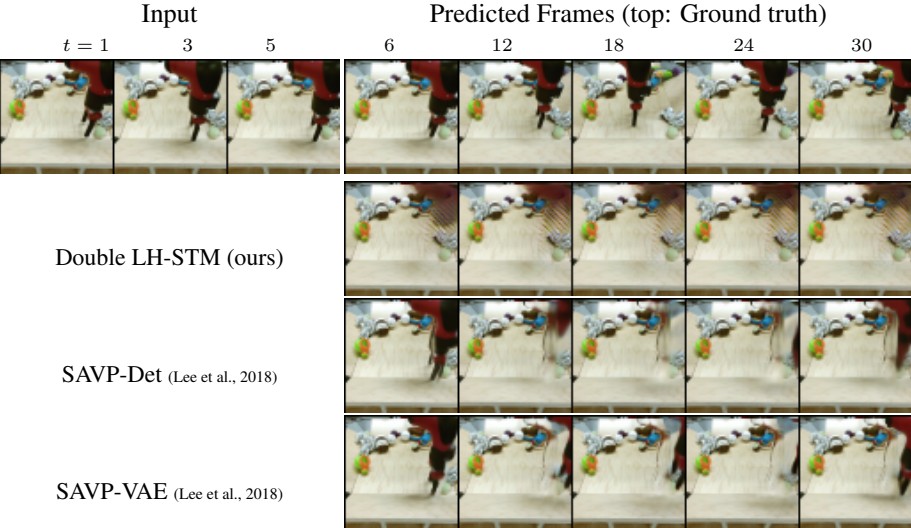

Figure 19: An example video with one of the largest motion on the BAIR robot pushing dataset. The motion magnitude is computed as averaged $\mathcal{L}_2$ norm between target frames following Villegas et al. (2017a). Top row shows the ground truth, and the rest is the prediction results. The model is trained for 10 frames and predicted for 25 frames given 5 frames. When motion is larger, all models fail to produce reasonable predictions.

