# OpenReview forum: "Long History Short-Term Memory for Long-Term Video Prediction"
_ICLR.cc/2020/Conference — Reject_

### Official Review · AnonReviewer2 · 2019-10-20
**Official Blind Review #2**

**Rating:** 3

**Review:**

This paper presents a new RNN unit based on ConvLSTM for long-term video prediction. The proposed method is technically correct but lacks enough originality. I tend to reject this paper due to the following three reasons:

1. The major novelty of this paper is the LH-STM unit, which applies a temporal attention approach to historical hidden states. This module is very similar to the Recall gate of the E3D-LSTM [Wang et al. 2018b]. Besides, the Double LH-STM looks like an incremental extension of the Single LH-STM. As mentioned, it is technically correct, and yet has limited novelty for an ICLR paper.

2. The authors mainly compared the proposed network with the Context-VP model in the experiments (Fig 5, Fig 8, and Table 3), which is not enough. As far as I know, there are other existing methods for long-term video prediction, e.g. [Denton et al. 2017].

3. Another problem of the experiments is that Lee et al. [2018] proposed a stochastic model for video prediction, but the authors only compared the LH-STM with its deterministic version.

4. In Figure 11, there is no significant improvement by using LH-STM. Also, the authors might include more compared models on the pushing dataset.

**Experience Assessment:**

I have published one or two papers in this area.

**Review Assessment: Checking Correctness Of Derivations And Theory:**

N/A

**Review Assessment: Checking Correctness Of Experiments:**

I carefully checked the experiments.

**Review Assessment: Thoroughness In Paper Reading:**

I read the paper at least twice and used my best judgement in assessing the paper.

---

> ### Author Response · Authors · 2019-11-07
> **Response to Reviewer #2**
>
> We appreciate the reviewer’s constructive feedback.
>
> 1. Novelty
> Despite similarities among various attention-based approaches, specific details of the attention mechanism can make models more expressive and easier to train. We believe the differences between our model and E3D-LSTM are crucial.
>
> In the attention mechanism of E3D-LSTM, the gate vector R_{k} (recall gate) is first computed with values in (0, 1) due to sigmoid. This recall vector is then compared to past cell states (unbounded) via dot product to compute the attention weights, which are finally applied to select the relevant past states.
> $\text{Attention}(R_{k}, C_{k-m:k-1}) = \text{Softmax}(R_{k} \cdot C_{k-m:k-1}) \cdot C_{k-m:k-1}$, where $R_{k}$ is the recall gate.
>
> In contrast, in our model, we first transform all hidden states H to H^tilda. The most recent transformed hidden state (H^tilda_{k-1}) is then compared to those in the past to compute the attention weights. We believe that this is better than E3D-LSTM because the vectors being compared are of the same 'type' (transformed hidden states) and range of values, making our attention mechanism more natural and intuitive.
> $\text{Attention}( \tilde{H}_{k-1}, \tilde{H}_{k-m:k-2} ) = \text{Softmax}(\tilde{H}_{k-1} \cdot \tilde{H}_{k-m:k-2}) \cdot \tilde{H}_{k-m:k-2}$, where $\tilde{H}$ indicates the transformed hidden state.
>
> The Double LH-LSTM is a very specific method of compressing accessible past information into a single state that significantly improves results without requiring more parameters. Thus it builds upon the Single LH-STM in a unique way.
>
> 2. Comparison with other existing methods for long-term prediction
> We provided the comparisons not only with Context-VP but also with SAVP-deterministic and -VAE [Lee et al., 2018] in Table 3 and with SAVP-deterministic in Figs 5 and 8.
> We did not directly compare with SVG-LP [Denton et al. 2017] as SAVP-VAE produces similar results to it (reported in [Lee et al., 2018] Section 4.4)
>
> We will respond to the rest of feedback soon.

---

### Official Review · AnonReviewer3 · 2019-10-22
**Official Blind Review #3**

**Rating:** 3

**Review:**

The paper proposes a type of recurrent neural network module called Long History Short-Term Memory (LH-STM) for longer-term video generation. This module can be used to replace ConvLSTMs in previously published video prediction models. It expands ConvLSTMs by adding a "previous history" term to the ConvLSTM equations that compute the IFO gates and the candidate new state. This history term corresponds to a linear combination of previous hidden states selected through a soft-attention mechanism. As such, it is not clear if there are significant differences between LH-STMs and previously proposed LSTMs with attention on previous hidden states. The authors propose recurrent units that include one or two History Selection (soft-attention) steps, called single LH-STM and double LH-STM respectively. The exact formulation of the double LH-STM is not clear from the paper.  The authors then propose to use models with LH-STM units for longer term video generation. They claim that LH-STM can better reduce error propagation and better model the complex dynamics of videos. To support the claims, they conduct empirical experiments where they show that the proposed model outperforms previous video prediction models on KTH (up to 80 frames) and the BAIR Push dataset (up to 25 frames).

Overall I believe there are serious flaws with the paper that prevent acceptance in its current form.

First, I believe the paper starts from the wrong assumption, namely that current video prediction models are limited by their capacity to limit the propagation of errors and to capture complex dynamics. Instead, it is well known that the main difficulty for longer term video prediction is to manage the increasing uncertainty in future outcomes. Stochastic models such as SVG-LP or SAVP are currently the state-of-the-art in video generation, with deterministic models not being able to generate more than a few non-blurry frames of video. While the authors mention that they do not focus on future uncertainty here, it is not clear how the proposed model helps to generate better longer-term videos when it does not deal with what actually makes long-term video generation difficult. In addition, it's misleading to claim that current models produce high quality generations for "only one or less than ten frames", especially without defining high quality. Models such as SVG [1] or SAVP[2] can produce non-blurry videos for 30-100 frames for the BAIR dataset, for example.

The experiments are missing 1) SVG as a baseline, 2) metrics that correlate with human perception such as LPIPS or FVD [3] and 3) qualitative samples that compare to stochastic models. Deterministic models can achieve very high PSNR/MSE/SSIM scores but produce very bad samples, as these scores are maximized by blurry predictions that conflate all possible future outcomes. This is highly apparent when looking at samples, and metrics that correlate better with human perception are usually better to compare video prediction methods. Comparisons to SAVP are found in Table 1 and 3 but there are no figures comparing samples from this model to the proposed model. The samples from the proposed model on the BAIR Push dataset for example (found in the appendix) are of significant lower quality than those reported from SAVP or SVG, and at the same time they are not longer-term than the predictions from these models. Consequently, the experimental section does not correctly assess how this model can generate better longer-term prediction than current models and it also does not give an accurate assessment of the model with respect to the current state-of-the-art.

To sum up, the paper does not adequately address how the proposed model allows for longer-term video generation. It is missing critical qualitative comparisons to state-of-the-art models such as SVG and it is unclear how the proposed model is different from a ConvLSTM with attention on previous hidden states.

[1] Stochastic Video Generation with a Learned Prior. E.Denton and R. Fergus. ICML 2018
[2] Stochastic Adversarial Video Prediction. Lee et al. Arxiv 2018
[3] Towards Accurate Generative Models of Video: A New Metric & Challenges. Unterthiner et al. Arxiv 2018


--- Post-discussion update ---
The authors have addressed a number of points raised by the reviewers and I'm raising my score to a weak reject from a reject. There are important remaining issues with the experimental section and the conclusions reached from their results, and therefore I still think the paper is below the acceptance bar.

**Experience Assessment:**

I have published one or two papers in this area.

**Review Assessment: Checking Correctness Of Derivations And Theory:**

N/A

**Review Assessment: Checking Correctness Of Experiments:**

I carefully checked the experiments.

**Review Assessment: Thoroughness In Paper Reading:**

I read the paper at least twice and used my best judgement in assessing the paper.

---

> ### Author Response · Authors · 2019-11-07
> **The importance of stochasticity for long-term video prediction**
>
> We agree that stochasticity is a crucial challenge and it is very important to find better ways of incorporating uncertainty into future prediction. However, we emphasize that it is not the only challenge we need to overcome. It is not settled that the network architectures currently in use are sufficiently powerful and efficient for long-term prediction. Regardless of whether the model produces stochastic or deterministic outputs, it needs to extract important information from spatio-temporal data and retain this information longer into the future efficiently. If it unables to do so, its uncertainty about the future will increase even if the future is completely predictable given the past. Therefore, developing better model architectures is still important.
>
> To support our claims, we point the reviewers to Fig. 4 of the SAVP paper [2]. Lee et al. [2] reported in the evaluation of the KTH dataset that even the predictions of SAVP-deterministic model do not degrade over time but can still be blurry. Having stochasticity in the model alone (SAVP-VAE) does not reduce blurriness in the predictions. Because of the ability of our model, we can produce sharper predictions compared to both SAVP-deterministic and -VAE.
>
> On the robot push dataset, the concern of stochasticity is valid due to random motion in the videos. It is true that the output samples in this dataset are blurrier than those reported from the SAVP paper [2]. However, please note that the stochastic model samples shown in the SAVP paper are selected based on the highest VGG cosine similarity score between the output and the ground truth. In general, stochastic models can be expected to perform better for such random motion. However as we noted earlier, stochastic models are also likely to benefit from more powerful architecture just like deterministic models do in our evaluation. This is an important avenue to be explored.
>
> We will add this important discussion about stochasticity to the introduction in our paper.

---

> > ### Comment · AnonReviewer3 · 2019-11-13
> > **About stochasticity**
> >
> > Deterministic models will only be able to model deterministic datasets. The KTH dataset is fairly deterministic with few different motions repeated cyclically and very few data, so deterministic models are able to generate decent samples from this dataset for ~30 frame horizons. In fact, stochastic models usually do not produce diverse samples in this dataset because there is not a lot of uncertainty.
> >
> > However the real world is not deterministic. From a toy video dataset (Stochastic Moving MNIST) to Atari games, the BAIR Push dataset, Cityscapes or Kinetics, in all these datasets the future is a stochastic function of the past. A model that does not model future uncertainty can't be used for longer term video prediction for most practical applications as it will quickly degrade as future uncertainty grows. If the authors believe otherwise, I propose a simple test - train your model on Stochastic Moving MNIST (this is an extremely simple dataset of moving MNIST digit that bounce in uncertain directions from the SVG paper and models can be trained in a few hours) and show predictions that do not degrade after a stochastic digit bounce.
> >
> > The proposed model is analyzed on KTH and BAIR dataset. On KTH it is comparable to previous models - the few samples shown on the paper do not look significantly better than other deterministic and stochastic models. However, on BAIR Push the samples from the proposed model look considerably worse than those of SAVP or SVG. Samples from those models can be found on their project pages or they can be generated with their code. Furthermore, comparing models with PSNR or SSIM in stochastic datasets has been shown to not correlate with human judgement (see FVD [3]).  I do not disagree that better architectures are important for video prediction, but a model that aims to solve 'longer term video prediction' can't ignore the most important difficulty - how to model increasing future uncertainty.
> >
> > SAVP website: https://alexlee-gk.github.io/video_prediction/
> > SVG website: https://sites.google.com/view/svglp/

---

> > > ### Author Response · Authors · 2019-11-14
> > > **Response about stochasticity**
> > >
> > > We believe that having a better architecture and dealing with model uncertainty are both important for video prediction, and both are unsolved problems. In this paper, we focus on the first part.
> > > We agree that the real world is not deterministic but not completely stochastic either. We do not claim that our model can solve the problem of unknown influence over time. Instead, our model extracts the information from the past and maintains consistency over time. We would like to show that this is also an important factor for long-term prediction. As mentioned earlier, we will add a discussion about this to the introduction, and will gladly remove any part of the paper that gives the wrong impression about the importance of stochasticity.

---

> ### Author Response · Authors · 2019-11-07
> **Rest of response to Reviewer #3**
>
> 1. "it is unclear how the proposed model is different from a ConvLSTM with attention on previous hidden states".
> ConvLSTM with attention on previous hidden states would be similar to our single LH-STM in time direction, but to our knowledge, there is no previous work that uses such a module. Could you provide any reference we have missed?
> In addition, we also proposed Double LH-STM which produces better results without requiring more parameters. This modification has not been explored in previous works with attention.
>
> 2. "SVG as a baseline"
> We did not directly compare with SVG since SAVP-VAE [2] produces similar results to it (reported in [2] Section 4.4).
>
> 3. "they are not longer-term than the predictions from these models."
> We followed a similar experimental setup to SAVP [2]. SAVP predicts 30 frames on KTH and 28 frames on BAIR. We reported 40 and 80 frames on KTH and 25 frames on BAIR.
>
> 4. "The exact formulation of the double LH-STM is not clear from the paper."
> Double LH-STM is divided into two blocks: History LSTM (H-LSTM) and Update LSTM (U-LSTM). The formulation of H-LSTM is shown in Eq 5. U-LSTM is the same as the standard LSTM (Eq 2) except the hidden state (H_{k-1}). The hidden state is replaced with the output of H-LSTM (H'_{k-1}). We will add the U-LSTM formulation in the paper.
>
> We will respond to the rest of the feedback as soon as possible.

---

> > ### Comment · AnonReviewer3 · 2019-11-13
> > **Response**
> >
> > 1 - [4] and [5], among other papers, use attention mechanisms for video prediction. In particular, [2] uses (separable) soft-attention for ConvLSTMs. Furthermore, since the ConvLSTM and LSTM formulation the same except that matrix multiplications are replaced by convolutions in ConvLSTM, the addition of an attention mechanism to LSTMs has been extensively explored in other applications (soft attention for language models/machine translation, image captioning, transformers, etc.).
> >
> > [4] https://arxiv.org/abs/1906.02634
> > [5] https://arxiv.org/abs/1907.06571
> >
> > 2 - There are differences between SVG-LP and SAVP-VAE. In particular, SAVP-VAE does not learn a prior and uses a significantly difference architecture.
> >
> > 3 - If you are replicating the setup, how does your model generate "longer term" predictions compared to current models as claimed in the paper? SVG and SAVP can also be unrolled for 100 frames on BAIR for example without extreme degradation (can be seen on their websites or using their code to generate samples).

---

> > > ### Author Response · Authors · 2019-11-14
> > > **Response to Reviewer #3**
> > >
> > > 1. The significant difference between our paper and others is: where the attention is used and what information is attended to.
> > >
> > > Attention modules in these papers soft-select video frames/sub-volumes directly. This process is independent from ConvLSTM. This mechanism is similar to 'input history' in our paper Section 4.2 (Fig. 4 left).
> > >
> > > In our paper, the attention soft-selects the hidden states inside of ConvLSTM. One of the major components in ConvLSTM/LSTM is a hidden state ($H_{k-1}$. The idea in our paper is that LSTM module can also use its own past information from the states $H_{k-m:k-2}$ for the time step $k$. The attention here is used to extract the past information by comparing it with the last state $H_{k-1}$ (see Fig. 1 (c) and Fig. 2).
> > >
> > > We believe that these states compress spatio-temporal information from the entire video better to select the relevant past information for video prediction. The impact of our proposed attention instead of using 'input' is shown in Tab. 1.
> > >
> > > The closest work we believe is [Cheng2016] using a standard LSTM for language models (referred in our paper, Section 2). We will add [4] and [5] as the relevant previous works as well.
> > >
> > > 2. We meant the performance comparison between SVG and SAVP-VAE are already reported in the SAVP paper (page 10-12 in their Arxiv paper) in terms of realism. Therefore, we did not report the numbers separately in this paper. We will refer SVG and their performance compared to SAVP-VAE in our paper.
> > >
> > > 3. We do not claim that our model can predict 'longer' then other papers, but learns long-term dependencies for videos. Other papers such as SAVP or SVG measured their performance up to 30-40 frames in their papers. We followed the same, then extended the evaluation to 80 frames. Only one figure in SVG shows some samples predicting 100 frames, and there are samples on the websites, but there are no systematic evaluations over the complete test set like we provide.
> > >
> > > The 'Longer prediction' in Section 4.3 refers longer than 40 frame prediction. We will correct the language in our paper, so the readers will not get the impression that our model can predict longer than others.

---

> > > > ### Comment · AnonReviewer3 · 2019-11-15
> > > > **Response**
> > > >
> > > > 1 - While it's true that the attention mechanisms proposed here and in the references differ in implementation, it is not clear to me what is the novelty and the effectiveness of the proposed method.
> > > >
> > > > In practice the method proposed here is very similar to standard soft-attention on previous hidden states of a LSTM. This is commonly done in the language community. The difference between the references and the proposed attention mechanism is basically the difference between transformers and LSTMs with attention. In the language community transformers currently outperform LSTM approaches. A comparison with such methods would help understand how this model stands in comparison to the references.
> > > >
> > > > 3 - Quote from the abstract: "our approach produces not only
> > > > sharper near-future predictions, but also farther into the future compared to the
> > > > state-of-the-art methods"
> > > >
> > > > Quote from the introduction: "However, recent stateof-the-art approaches produce high-quality predictions only for one or less then ten frames"
> > > >
> > > > These are misleading claims and directly imply that the proposed model can produce longer term videos than current methods.

---

> > > > > ### Author Response · Authors · 2019-11-15
> > > > > **Response to Reviewer #3**
> > > > >
> > > > > The effectiveness of our paper compared to attention over inputs instead of states are shown in Section 4.2.
> > > > > It is true that attention on LSTM in language modeling is popular, however this has not bee explored in video prediction. Considering such high-dimensional spatio-temporal data has different properties, we believe that our contribution is important for the community.
> > > > > Thanks for the suggestions. We will consider them in the next update.

---

### Official Review · AnonReviewer1 · 2019-10-23
**Official Blind Review #1**

**Rating:** 3

**Review:**

Summary:
This paper proposes a new LSTM architecture called LH-STM (and Double LH-STM). The main idea deals with having a history selection mechanism to directly extract what information from the past. The authors also propose to decompose the history and update in LH-STM into two networks called Double LH-STM. In experiments, the authors evaluate and compare their two architectures with previously proposed models. They show that their architecture outperforms previous in the PSNR, SSIM and VIF metrics.


Pros:
+ New architecture that can use all computed state history in a sequence
+ Outperforms previous methods in the used metrics

Weaknesses / comments:
- Larger number of parameters in comparison to ContextVP-4
In Table 1, the authors present a comparison to previous works and the respective number of parameters used for most of the methods. It is mentioned in the paper that Single LH-STM uses an architecture similar to ContextVP-4. Since the architectures are similar and the proposed method is added to this architecture, can the authors make sure that Context VP-4 and Single LH-STM have around the same number of parameters? This can give a more direct comparison in performance to make sure that the parameter boost is not the reason for performance boost.


- Why retrain for longer sequences?
The authors have experiments where they attempt to predict past 40 frames into the future. However, they mention that for this experiment, they train another network that takes in 64x64 pixels. Optimally, they should just let the 128x128 network predict past 40 frames. Can the authors comment on why they don’t just do this? Is long-term prediction limited to how many previous states you can store in the GPU?


- PSNR, SSIM, and VIF could be biased to blurriness and perfect background reconstruction
It has been shown before that PSNR and SSIM can be biased to blurriness and perfectly copying the background (Villegas et al., 2017b). Therefore, these metrics should be complemented with other metrics such as actual humans look at the videos and evaluated them for realism. The fact that these metrics look good in this paper can be due to blurriness. There is clear evidence of blurry predictions in the comparison Figures in the main paper and supplementary material. In addition these metrics, and VIF can also be biased to perfectly copying the background. I suggest the authors separate the data into videos with large motion and little motion similar to Villegas et al., 2017a. This way we can better evaluate this method on how well it predicts videos with large motion in comparison with videos where copying the background is enough.


- Testing for 80 frame sequences is not very meaningful in this dataset.
The KTH dataset contains the action categories of running, walking, and jogging which most of these videos do not go up up 40 frames while the person still being in the frame. Therefore, after frame 40, the method is just required to copy the background and it will look like the future is perfectly predicted.. Can the authors clarify if they only tested on handwave, handclap and boxing? Handwave, handclap and boxing are the only categories that will still have a human present in the video at frame 80.


- No video files are provided.
Finally, this method does not provide any videos to better evaluate the proposed network. Looking at videos is necessary to observe temporal consistency and blurriness happening in the video. Or humans appearing and disappearing randomly.


Conclusion:
The proposed method is novel and interesting, but the experimental section has many issues as discussed above. If the authors successfully address these issues, I am willing to increase my score.


**Experience Assessment:**

I have published in this field for several years.

**Review Assessment: Checking Correctness Of Derivations And Theory:**

N/A

**Review Assessment: Checking Correctness Of Experiments:**

I carefully checked the experiments.

**Review Assessment: Thoroughness In Paper Reading:**

I read the paper thoroughly.

---

> ### Author Response · Authors · 2019-11-07
> **Response to Reviewer #1**
>
> We thank the reviewer for constructive comments.
>
> 1. "Larger number of parameters in comparison to ContextVP-4"
> Both Single and Double LH-STM use ContextVP as the base model.
> We adjusted the number of neurons, so all models have similar model size (≈17M) (stated in Appendix A).
> Due to the different design of the model, we could not make the exact same number of parameters (Single is 7.5% larger than ContextVP).
> To follow the rule of thumb for GPU parallelization, we kept the layer size divisible by 32.
> Double LH-STM has the smallest number of parameters (6.3% less than ContextVP) but produces the best performance. This indicates that the performance boost is not due to the parameters.
>
> 2. Why retrain for longer sequences?
> Due to lack of GPU memory, we could not use the original resolution for 80 frame prediction.
>
> We will respond to the rest of the comments as soon as possible.

---

> > ### Author Response · Authors · 2019-11-15
> > **video**
> >
> > We have uploaded a GIF showing sample comparisons of 7 videos to an anonymous sharing website.
> > Link: https://postimg.cc/nCC5k2nc
> > The first row is the ground truth.
> > The second, third, and last row show outputs of Double LH-STM, SAVP-Deterministic, and SAVP-VAE, respectively.
> > Each column corresponds to a separate test set video sample.
> > We hope this addresses your concern.

---

### Author Response · Authors · 2019-11-13
**Paper updated with requested results and equations**

We have updated the paper with more results (see Fig. 5-8 and 10-19).

1. SAVP-VAE results have been included for all comparisons. [reviewer 2 and 3]

2. LPIPS metric has been added  for all additional results. [reviewer 3]

3. Separating the data with different action classes. (Fig 5, 6 and Fig. 12, 13) [reviewer 1]
In addition to the results with all actions (Fig. 5), we also show per-frame results with only 'boxing', 'waving', and 'clapping' (Fig. 6) action classes for 80 frame prediction. Double LH-STM results are generally better than SAVP-Deterministic and -VAE on all metrics. This shows that our model is not just copying the background when predicting more than 40 frames. We additionally provide per-frame comparisons for each action class in Fig. 12-13.

4. Separating the data with large and small motion similar to Villegas et al., 2017a. (Fig. 10, 11) [reviewer 1]
We split the video samples into five bins by motion magnitude computed as averaged L2 norm between target frames. Overall, across all metrics, all models perform worse when the motion is larger. For PSNR and SSIM, Double LH-STM achieves the best performance except for the largest motion. For LPIPS, Double LH-STM performs better for small motions (first two bins). SAVP-Deterministic performs the best on this metric for larger motions.

5. More samples for both datasets. (Fig. 8, 14-16 for the KTH action dataset and Fig. 17-19 for the robot push dataset) [reviewer 1,2,3]
On KTH action dataset, samples in Fig. 8, 14 and 15 show that motion and human shape predicted by Double LH-STM are the closest to the ground truth. On robot push dataset, random robot motion makes long-term prediction extremely hard with deterministic models as reviewer 3 pointed out. However, the overall motion prediction with Double LH-STM and SAVP-Deterministic is still reasonable compared to SAVP-VAE. We also included the samples with one of the largest motions for both datasets (Fig 16 and 19). When motion is larger, all models fail to produce reasonable predictions, whether they are deterministic or stochastic.

6. The exact formulation of the double LH-STM has been added. (Eq. 5, 6) [reviewer 3]

This update only includes extra Figures and Equations requested by reviewers, but we have not updated the main text yet. We will update it soon.
We welcome additional suggestions or questions.

---

> ### Comment · AnonReviewer3 · 2019-11-13
> **LPIPS metric**
>
> Could the authors explain how is the LPIPS metric computed in the new plots? In particular, which sample(s) from SAVP-VAE is/are used to compare to the deterministic models?
>
> In general SAVP-VAE obtains much better LPIPS scores than SAVP-Deterministic. These can be seen in Figure 14 (BAIR) and Figure 15 (for KTH) of the SAVP submission to ICLR 2019 ( https://openreview.net/pdf?id=HyEl3o05Fm ). I have been able to replicate the results on these figures with the official SAVP code, so I am wondering how the authors obtained the plots in Figure 11 for BAIR or the other figures for KTH in which SAVP-Deterministic consistenly outperforms SAVP-VAE? Just by looking at the added generations of SAVP-VAE compared to those of SAVP-Deterministic in the appendix it can be seen that they are significantly better, and the plots suggest otherwise.

---

> > ### Author Response · Authors · 2019-11-14
> > **About SAVP-VAE evaluation with LPIPS metric**
> >
> > SAVP-VAE numbers in their paper are obtained by first generating 100 output samples per input and then selecting the best generated sample for each input based on the highest score between the output and the ground truth. This evaluation measures the best prediction the model *can* do, given 100 tries. However, it does not provide an overall picture of the generations from the model. In real world scenarios, the ground truth is not available, so the 'best' samples cannot be picked as done in such evaluations. Therefore, we re-computed the scores for stochastic models based on their 'median' output sample (instead of best) among the 100 randomly generated ones, compared to the ground truth (stated in our paper, page 6 footnote). This strategy measures how well the model can be expected to perform on average, and so it is a more representative score. We will update the paper with a detailed discussion of our evaluation.

---

> > > ### Comment · AnonReviewer3 · 2019-11-15
> > > **Misleading experiment**
> > >
> > > Taking the mean or the median of different predictions for the stochastic model is unfair as you are averaging predictions producing blurry results The advantage of stochastic methods is to precisely avoid these average blurry predictions of all possible future outcomes.
> > >
> > > Furthermore you do have a mechanism to select at least a greedily best sample from SVG/SAVP-VAE by sampling the latent variable with the highest probability at each timestep, i.e. sample the mean of the gaussian probability for the latent variable at a given timestep.

---

> > > > ### Author Response · Authors · 2019-11-15
> > > > **Misunderstanding**
> > > >
> > > > There appears to have been a misunderstanding. We did not average the predictions. Our reported numbers are the median score out of the scores of all 100 predictions.
> > > > If one out of many samples produce a perfect prediction, it is not really useful in reality. High median score means the model will more likely produce good prediction on average.
> > > > The next suggestion is a good idea. We will consider this evaluation in the next update. Nevertheless, we believe that our current metric is more useful than the comparison based on best samples in previous work.

---

### Author Response · Authors · 2019-11-15
**paper updated**

We have now updated the main text of the paper as well based on all comments from the reviewers.  In particular, we have added a discussion about stochasticity to the introduction that clarifies our focus.

---

### Decision · Program_Chairs · 2019-12-19

**Decision:**

Reject

**Comment:**

The paper proposes a new recurrent unit which incorporates long history states to learn longer range dependencies for improved video prediction. This history term corresponds to a linear combination of previous hidden states selected through a soft-attention mechanism and can be directly added to ConvLSTM equations that compute the IFO gates and the new state. The authors perform empirical validation on the challenging KTH and BAIR Push datasets and show that their architecture outperforms existing work in terms of SSIM, PSNR, and VIF.
The main issue raised by the reviewers is the incremental nature of the work and issues in the empirical evaluation which do not support the main claims in the paper. After the rebuttal and discussion phase the reviewers agree that these issues were not adequately resolved and the work doesn’t meet the acceptance bar. I will hence recommend the rejection of this paper. Nevertheless, we encourage the authors improve the manuscript by addressing the remaining issues in the empirical evaluation.